# SIMDIFF: SIMULATOR-CONSTRAINED DIFFUSION MODEL FOR PHYSICALLY PLAUSIBLE MOTION GENERATION

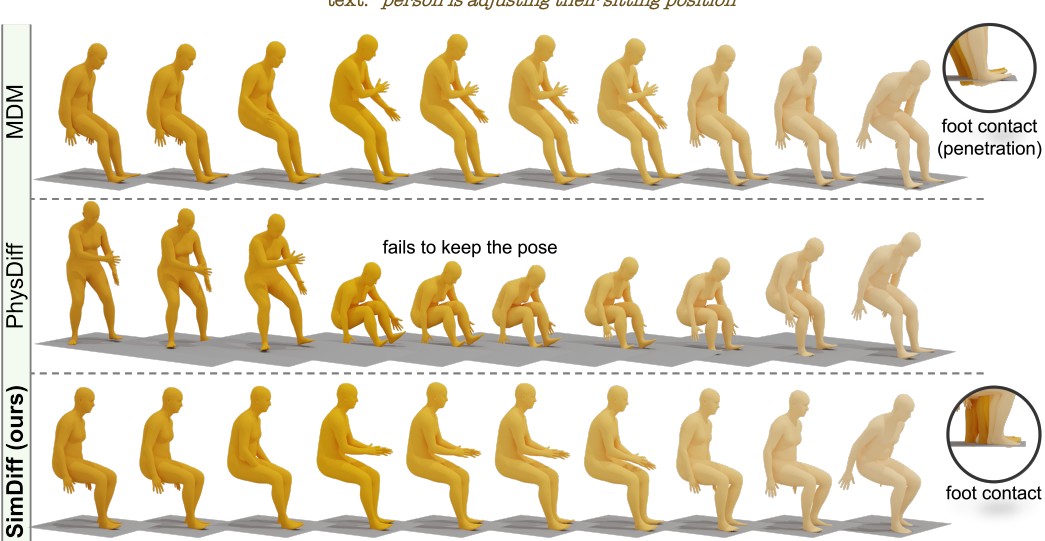

Figure 1: Our SimDiff generates physically plausible motions by training only lightweight adapters on simulator-augmented data, eliminating artifacts such as foot–ground penetration and pose instability.

## ABSTRACT

Generating physically plausible human motion is crucial for applications such as character animation and virtual reality. Existing approaches often incorporate a simulator-based motion projection layer to the diffusion process to enforce physical plausibility. However, such methods are computationally expensive due to the sequential nature of the simulator, which prevents parallelization. We show that simulator-based motion projection can be interpreted as a form of guidance—either classifier-based or classifier-free—within the diffusion process. Building on this insight, we propose **SimDiff**, a Simulator-constrained Diffusion Model that integrates environment parameters (e.g., gravity, wind) directly into the denoising process. By conditioning on these parameters, SimDiff generates physically plausible motions efficiently, without repeated simulator calls at inference, and also provides fine-grained control over different physical coefficients. Moreover, SimDiff successfully generalizes to unseen combinations of environmental parameters, demonstrating compositional generalization.

## 1 INTRODUCTION

Believable motions shape how the audience perceives a character's personality and surroundings. Motion generation task refers to automatically producing realistic character motions under various specified conditions, such as textual prompts. This process can reduce the need for extensive manual work, offering animators a more efficient way to develop diverse character animations.

Diffusion models (Tevet et al., 2023; Zhang et al., 2022; 2023b) have recently emerged as promising approaches for generating a wide variety of human motions by learning from large-scale datasets such as HumanML3D (Guo et al., 2022). These models effectively capture the multimodality of human motion, aided by text annotations and other rich contexts such as partial keyframe constraints. However, the motion data collected via standard motion capture is not always physically plausible, as it can contain artifacts like slight floating or inaccurate foot contacts. Additionally, since such datasets typically consist of motions captured under uniform conditions (e.g., Earth gravity and zero wind), these models have no direct knowledge of how to generate physically plausible motions in different environments, such as the low gravity on the Moon, where the same movements would behave differently.

Recent work has explored combining diffusion models with physics simulators to generate physically plausible motions (Yuan et al., 2023; Ren et al., 2023; Gillman et al., 2024). PhysDiff (Yuan et al., 2023) employs a physics-based projection at specific diffusion steps to correct denoised samples during inference, and InsActor (Ren et al., 2023) applies a simulator-driven post-processing step at inference time. By contrast, Gillman et al. (Gillman et al., 2024) proposed a self-correcting loop that fine-tunes a motion generative model with simulated data, iteratively refining its outputs. However, these approaches have not yet provided (i) a standalone physics-aware text-to-motion generative model that internalises simulator behaviour, eliminating the need for any simulator or controller at test time, nor (ii) a discussion of how to extend their methods to unseen environments.

We therefore present a Simulator-constrained Diffusion Model (**SimDiff**), which directly incorporates physical constraints into the diffusion process. Drawing inspiration from how humans can often judge motion plausibility without exhaustively simulating every physical detail, SimDiff employs classifier-free guidance (Ho & Salimans, 2022) with an implicit classifier to steer the denoising process toward physically plausible motions. Environment parameters, such as gravity and wind conditions, are used as conditional signals, which we efficiently inject by training only a small set of lightweight adapters added to a frozen diffusion backbone. With these signals, SimDiff generates motions that respect environment-specific constraints without relying on external simulators or post-processing.

Moreover, we show that the motion projection process in PhysDiff (Yuan et al., 2023) can be understood within the framework of classifier guidance (Dhariwal & Nichol, 2021; Nichol et al., 2021). Building on this perspective, we argue that integrating physical constraints directly into the diffusion process as a condition provides a more unified and theoretically grounded approach.

To summarize, the main contributions of this work are as follows

- **Simulator-Constrained Diffusion Model**: We propose SimDiff, a simulator-constrained diffusion model that integrates physical constraints directly into the diffusion process using classifier-free guidance, enabling the generation of physically plausible human motions without the need for simulation during inference.
- **Reinterpretation of PhysDiff**: We provide a theoretical explanation of PhysDiff from the perspective of traditional methods of conditioning diffusion models by clarifying the classifier that PhysDiff assumes. This theoretical insight allows us to extend our approach to generate motions across diverse environments by conditioning on physical parameters.
- **Adaptability to Diverse Conditions**: By conditioning on explicit physical parameters, SimDiff can flexibly generate motions in various environments without retraining specialized controllers.

## 2 RELATED WORK

### 2.1 DIFFUSION MODELS FOR HUMAN MOTION GENERATION

Diffusion models have recently emerged as a powerful class of neural generative models, demonstrating significant advancements in content creation across various domains, including image synthesis (Dhariwal & Nichol, 2021; Ramesh et al., 2022; Rombach et al., 2022; Saharia et al., 2022), video synthesis (Ho et al., 2022; Wu et al., 2023; Blattmann et al., 2023), and text-to-speech synthesis (Kong et al., 2021; Popov et al., 2021). These models generate data by reversing a diffusion process that progressively adds noise to data samples, enabling them to produce high-quality and diverse outputs through denoising. In the domain of human motion generation, diffusion models have

shown promising results (Tevet et al., 2023; Zhang et al., 2022; 2023b), outperforming traditional methods based on autoencoders (Yan et al., 2018; Aliakbarian et al., 2020), Variational Autoencoders (VAEs), Generative Adversarial Networks (GANs) (Barsoum et al., 2017; Harvey et al., 2020; Wang et al., 2020), and Normalizing Flow Networks (Henter et al., 2020). Building on these successes, our work specifically targets diffusion models.

## 2.2 INTEGRATING PHYSICS-BASED METHODS INTO MOTION DIFFUSION MODELS

Physics-based character animation techniques can generate complex and physically plausible motions by training imitation policies using reinforcement learning (RL) (Peters & Schaal, 2008; Sutton & Barto, 2018; Peng et al., 2018; 2021; 2022; Zhang et al., 2023a; Tirinzoni et al., 2025). By learning motor skills through RL in physics simulators that enforce physical laws, these methods ensure that the resulting motions inherently obey those laws. To improve the physical plausibility of motions generated by diffusion models, previous work has explored incorporating these physics-based methods. PhysDiff (Yuan et al., 2023) replaces the diffusion model's outputs at certain time steps with physically plausible motions obtained through physics-based methods. Concurrently, Trace & Pace (Rempe et al., 2023) couples a guided trajectory-diffusion generator with a physics-based pedestrian controller, yielding user-controllable yet physically grounded animations. InsActor (Ren et al., 2023) employs a hierarchical framework that leverages a controller to refine motion transitions, effectively mimicking a high-level diffusion planner. Gillman et al. (2024) introduced a self-correcting loop that fine-tunes a motion generative model by correcting its intermediate outputs with physics-based methods and reusing the adjusted motions for further training. RobotMDM (Serifi et al., 2024) and ReinDiffuse (Han et al., 2025) aimed to internalize physics constraints by fine-tuning diffusion models using reinforcement learning.

## 3 PRELIMINARIES

**Motion Representation.** We use two different motion representations, each suitable for its purpose. For the kinematic motions generated by the diffusion model, we follow MDM (Tevet et al., 2023) and use the HumanML3D (Guo et al., 2022) format, where every frame is stored relative to the previous one. For the RL tracking policy, we adopt the SMPL humanoid model (Loper et al., 2015), widely used in virtual character animation (Yuan et al., 2021; Luo et al., 2023; 2024; Tirinzoni et al., 2025). The SMPL skeleton consists of 24 rigid bodies, of which 23 are actuated, with states containing body pose (70D), body rotation (144D), and linear and angular velocities (144D), resulting in a 358-dimensional state. To convert HumanML3D to SMPL, we fit SMPL joint rotations and root positions to the HumanML3D trajectories using SMPLify (Bogo et al., 2016), then compute velocities via finite differences. The inverse conversion reconstructs relative root translations and rotations by differentiating absolute positions and rotations obtained through forward kinematics. For brevity, we use the same symbols $\tau$ and $\mathbf{s}$ to represent motion sequences and states, respectively, across both representations.

**Physics-projection module.** Let $\mathcal{P}_{\phi,\pi}$ be a physics-based projection operator that maps a kinematic motion sequence $\tau$ to a physically plausible rollout $\hat{\tau} = \mathcal{P}_{\phi,\pi}(\tau)$. Here, $\phi$ denotes environment parameters (e.g., gravity, wind), and $\pi(\mathbf{a}|\mathbf{s})$ is an imitation policy producing proportional derivative (PD) controller targets at $30\,\mathrm{Hz}$. To distinguish simulation timesteps from diffusion steps $t$, we index simulation time by $n = 0, \ldots, N$. At each step $n$, the policy observes the current state $\mathbf{s}^n$ and outputs an action $\mathbf{a}^n$, which is transformed by a low-level PD controller into joint torques applied to the SMPL humanoid in MuJoCo (Todorov et al., 2012). The simulator advances at $450\,\mathrm{Hz}$ to produce the next physically plausible state $\hat{\mathbf{s}}^{n+1}$. Iterating this process yields a physically plausible motion sequence $\hat{\tau} = \{\hat{\mathbf{s}}^0, \ldots, \hat{\mathbf{s}}^N\}$ that closely tracks the original motion while satisfying physical constraints.

**Diffusion Models.** Diffusion models are a class of generative models that learn to gradually denoise a sample that has been noised by a forward diffusion process (Ho et al., 2020). Let $\tau_0$ represent the original motion data, and $\tau_1, \ldots, \tau_T$ be the sequence of increasingly noisy versions of the data, where $T$ is the total number of diffusion steps. The forward process is defined as a Markov chain that

gradually adds Gaussian noise to the data over $T$ timesteps:

$$q(\boldsymbol{\tau}_t|\boldsymbol{\tau}_{t-1}) = \mathcal{N}\left(\boldsymbol{\tau}_t; \sqrt{1-\beta_t}\boldsymbol{\tau}_{t-1}, \beta_t\boldsymbol{I}\right) \tag{1}$$

where $q(\boldsymbol{\tau}_t|\boldsymbol{\tau}_{t-1})$ is the transition probability from $\boldsymbol{\tau}_{t-1}$ to $\boldsymbol{\tau}_t$, $\beta_t \in (0,1)$ is a variance schedule, $\mathcal{N}(\mu, \sigma^2)$ denotes a Gaussian distribution with mean $\mu$ and variance $\sigma^2$, and $\boldsymbol{I}$ is the identity matrix.

The reverse process, learned by the model, gradually denoises the sample:

$$p_\theta(\boldsymbol{\tau}_{t-1}|\boldsymbol{\tau}_t) = \mathcal{N}\left(\boldsymbol{\tau}_{t-1}; \boldsymbol{\mu}_\theta(\boldsymbol{\tau}_t, t), \Sigma_\theta(\boldsymbol{\tau}_t, t)\right) \tag{2}$$

where $p_\theta(\boldsymbol{\tau}_{t-1}|\boldsymbol{\tau}_t)$ is the learned reverse transition probability, $\theta$ represents the parameters of the model, $\boldsymbol{\mu}_\theta(\boldsymbol{\tau}_t, t)$ is the predicted mean, and $\Sigma_\theta(\boldsymbol{\tau}_t, t)$ is the predicted covariance matrix.

The model is trained to predict the noise $\boldsymbol{\epsilon}$ added at each step, which can be used to estimate the mean of the reverse process:

$$\boldsymbol{\mu}_\theta(\boldsymbol{\tau}_t, t) = \frac{\sqrt{\alpha_t}(1-\bar{\alpha}_{t-1})}{1-\bar{\alpha}_t}\boldsymbol{\tau}_t + \frac{\sqrt{\bar{\alpha}_{t-1}}\beta_t}{1-\bar{\alpha}_t}\boldsymbol{\tau}_0 = \frac{1}{\sqrt{1-\beta_t}}\left(\boldsymbol{\tau}_t - \frac{\beta_t}{\sqrt{1-\bar{\alpha}_t}}\boldsymbol{\epsilon}_\theta(\boldsymbol{\tau}_t, t)\right) \tag{3}$$

where $\bar{\alpha}_t = \prod_{s=1}^{t}(1-\beta_s)$ and $\boldsymbol{\epsilon}_\theta(\boldsymbol{\tau}_t, t)$ is the predicted noise.

During sampling, the model starts from pure noise $\boldsymbol{\tau}_T \sim \mathcal{N}(\boldsymbol{0}, \boldsymbol{I})$ and iteratively denoises it to generate a sample from the learned data distribution (Sohl-Dickstein et al., 2015a).

## 4 METHOD

We begin by formulating physically plausible motion generation within a classifier-guided diffusion framework, where a classifier indicates whether a given motion is physically plausible. From this perspective, PhysDiff (Yuan et al., 2023) can be viewed as guiding the denoising process to minimize the difference between the generated motion and a physically plausible reference motion. Building on this idea, we propose **SimDiff**, a Simulator-Constrained Diffusion Model for physically plausible motion generation, which integrates environment-specific constraints directly into the diffusion process. Rather than relying on an explicit classifier or post-processing steps, SimDiff learns these constraints from simulated data under diverse conditions, enabling it to generate motion trajectories that respect physical principles without requiring external simulator corrections at inference time.

### 4.1 SIMULATOR-CONSTRAINED DIFFUSION MODEL

We aim to define a distribution from which physically plausible motions can be sampled. In traditional classifier-guided diffusion models, this amounts to considering

$$p(\boldsymbol{\tau}|\mathcal{Y}=1) \propto p(\boldsymbol{\tau})p(\mathcal{Y}=1|\boldsymbol{\tau}), \tag{4}$$

where $\mathcal{Y}$ is a binary random variable, with $\mathcal{Y}=1$ indicating that the trajectory of motion data $\boldsymbol{\tau}$ is physically plausible.

To define the likelihood $p(\mathcal{Y}=1|\boldsymbol{\tau}_t)$ that a motion at time step $t$ is physically plausible, we assume the existence of a clean, physically plausible motion $\hat{\boldsymbol{\tau}}_0$ from which the plausibility of noisy motions can be evaluated. We now assume this likelihood can be expressed as

$$p(\mathcal{Y}=1|\boldsymbol{\tau}_t) := \exp\left(-\|\boldsymbol{\tau}_t - \hat{\boldsymbol{\tau}}_t\|^2\right), \tag{5}$$

where $\hat{\boldsymbol{\tau}}_t = \sqrt{\bar{\alpha}_t}\hat{\boldsymbol{\tau}}_0 + \sqrt{1-\bar{\alpha}_t}\boldsymbol{\epsilon}$ is a motion transformed back to time $t$ from the physically plausible motion $\hat{\boldsymbol{\tau}}_0$ with the addition of scheduled *i.i.d* gaussian noise $\boldsymbol{\epsilon} \sim \mathcal{N}(\boldsymbol{0}, \boldsymbol{I})$. This classifier assigns a high likelihood to motions $\boldsymbol{\tau}_t$ that closely align with the physically plausible motion at time $t$.

When the conditional probability $p(\mathcal{Y}=1|\boldsymbol{\tau}_t)$ is sufficiently smooth, the transitions in the reverse diffusion process can be approximated as Gaussian (Sohl-Dickstein et al., 2015b)

$$p(\boldsymbol{\tau}_t|\boldsymbol{\tau}_{t+1}, \mathcal{Y}=1) \approx \mathcal{N}(\boldsymbol{\tau}_t; \boldsymbol{\mu} + \gamma\Sigma\boldsymbol{g}, \Sigma) \xrightarrow{\text{PhysDiff}} \mathcal{N}(\boldsymbol{\tau}_t; \hat{\boldsymbol{\tau}}_t, \Sigma), \tag{6}$$

where $\boldsymbol{\mu}$ and $\Sigma$ are parameters from the original reverse process transition $p_\theta(\boldsymbol{\tau}_t|\boldsymbol{\tau}_{t+1})$. The gradient $\boldsymbol{g}$ can be computed as

$$\boldsymbol{g} = \nabla_{\boldsymbol{\tau}_t}\log p(\mathcal{Y}=1|\boldsymbol{\tau}_t)|_{\boldsymbol{\tau}_t=\boldsymbol{\mu}} = -2\left(\boldsymbol{\mu} - \hat{\boldsymbol{\tau}}_t\right). \tag{7}$$

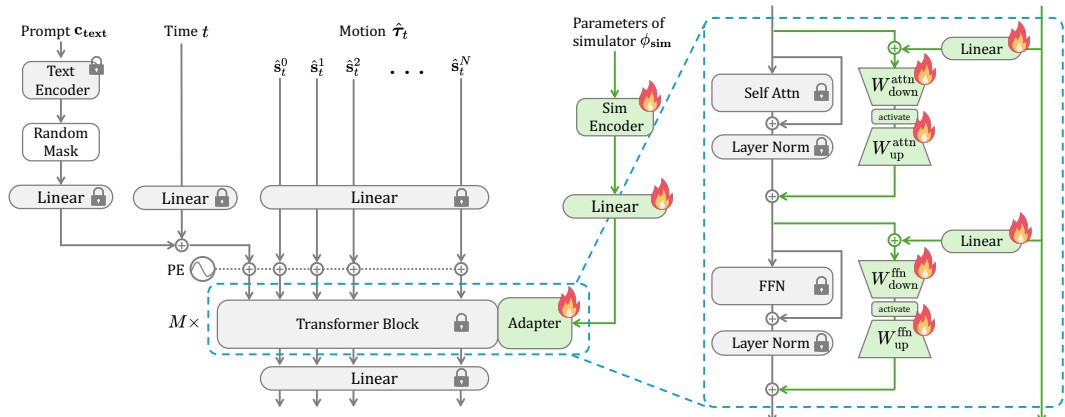

Figure 2: **SimDiff overview.** A frozen MDM backbone (grey boxes with 🔒) processes the text prompt $c_{text}$, the diffusion timestep $t$, and the partially-noised, physically plausible motion sequence $\hat{\tau}_t$. Simulator parameters $\phi_{sim}$ are embedded by a trainable Sim Encoder, producing an environment embedding. This embedding is injected into the model through lightweight Motion Adapters, which are inserted in parallel to every Transformer layer. Only the green modules marked with 🔥 are trained.

By selecting $\gamma$ and $\mathbf{\Sigma}$ such that $\gamma g \mathbf{\Sigma} = g/2$, this results in

$$\boldsymbol{\mu} + \gamma g \mathbf{\Sigma} = \hat{\boldsymbol{\tau}}_t. \tag{8}$$

This equation shows that the mean value of the original model is replaced by the physically plausible motion at time step $t$.

In previous work, PhysDiff (Yuan et al., 2023) obtains the clean, physically plausible motion $\hat{\tau}_0$ using a physics-based motion projection module $\mathcal{P}_\pi$, which consists of an imitation policy and a physics simulator. Within the context of VP-SDE or DDPM sampling, PhysDiff estimates the posterior mean from $p(\boldsymbol{\tau}_0|\boldsymbol{\tau}_t)$ by applying Tweedie's approach (Chung et al., 2023; Efron, 2011; Kim & Ye, 2021) as

$$\hat{\boldsymbol{\tau}}_t = \frac{\sqrt{\alpha_t}(1 - \bar{\alpha}_{t-1})}{1 - \bar{\alpha}_t}\boldsymbol{\tau}_t + \frac{\sqrt{\bar{\alpha}_{t-1}}\beta_t}{1 - \bar{\alpha}_t}\mathcal{P}_\pi(\tilde{\boldsymbol{\tau}}_0), \tag{9}$$

where $\tilde{\boldsymbol{\tau}}_0 = \frac{1}{\sqrt{\bar{\alpha}_t}}\left(\boldsymbol{\tau}_t - \sqrt{1 - \bar{\alpha}_t}\,\boldsymbol{\epsilon}_\theta(\boldsymbol{\tau}_t, t)\right)$. PhysDiff can now be seen as an approximate realization of the conditional distribution in Equation (4). By repeatedly substituting a simulator-corrected motion into the denoising step, PhysDiff effectively pushes the sampled trajectory toward regions of motion space that satisfy physical constraints, approximating the conditional distribution $p(\boldsymbol{\tau} \mid \mathcal{Y} = 1)$.

However, repeatedly substituting a simulator-corrected reference at each step is computationally expensive, making it infeasible to apply guidance across all time steps. Therefore, we focus on directly learning the conditional distribution $p(\boldsymbol{\tau} \mid \mathcal{Y})$ from data. Instead of relying on inference-time substitutions, we train our model on simulator-generated data using classifier-free guidance (Ho & Salimans, 2022).

## 4.2 SIMULATOR-CONSTRAINED DIFFUSION MODEL FOR DIVERSE ENVIRONMENTS

While the binary concept of physical plausibility provides a foundation, physical plausibility itself is inherently dependent on environmental parameters such as gravity and friction. To account for this, we extend our model to condition on these simulator parameters, denoted as $\phi_{sim}$. This conditioning allows the model to generate motions that are physically plausible within the specific context of a given environment.

Learning the conditional distribution of motions under varying physical conditions requires motion data collected across diverse environments. However, since it is infeasible to gather real-world motion data for such a wide range of scenarios, we rely on a physics simulator to generate this data. This simulated data allows the model to learn the underlying relationships between environmental parameters and physically plausible motion patterns. We then aim to learn $p(\boldsymbol{\tau}|\phi_{sim})$ from this simulated data.

**Architecture:** SimDiff extends the pretrained MDM backbone (Tevet et al., 2023) by introducing Motion Adapters, which inject environment parameters into the model. Importantly, we leave the core diffusion components of MDM frozen and only train these lightweight adapters, thus preserving the original generative behavior while enabling environment-specific conditioning. At a high level, a small Sim Encoder first embeds the simulator parameters $\phi_{\text{sim}}$ into a vector $e_{\text{sim}}$. Then, at each Transformer (Vaswani et al., 2017) layer, a Motion Adapter uses $e_{\text{sim}}$ to steer the hidden features toward environment-specific motion.

Each adapter is placed in parallel with every residual branch of the Transformer (see the right-hand side of Fig. 2). Let $h_m$ be the hidden vector at layer $m$, $e_{\text{sim}} \in \mathbb{R}^d$ the environment embedding produced by the Sim Encoder, $\mathbf{W}_{\text{down}} \in \mathbb{R}^{r \times d}$ and $\mathbf{W}_{\text{up}} \in \mathbb{R}^{d \times r}$ bottleneck down- and up-projection matrices respectively (with $r < d$), and $\mathbf{W}_{\text{sim}} \in \mathbb{R}^{d \times d}$ a learnable linear layer mapping the environment embedding to the Transformer hidden dimension. The adapter refines $h_m$ as:

$$h'_m = h_m + \alpha \cdot \mathbf{W}_{\text{up}}\, \sigma\Big(\mathbf{W}_{\text{down}}\left(h_m + e_{\text{sim}}\, \mathbf{W}_{\text{sim}}\right)\Big), \tag{10}$$

where $\sigma$ denotes the SiLU activation and $\alpha$ controls the adapter's influence at inference (set to 1 during training, adjustable at inference). We zero-initialize $\mathbf{W}_{\text{up}}$ to ensure the pretrained MDM behavior is preserved initially and gradually guided by $e_{\text{sim}}$ during training.

**Training Data Generation**: We build a simulator-augmented corpus covering diverse physical conditions by replaying reference motions in MuJoCo (Todorov et al., 2012) with domain-randomized simulator parameters. For each motion clip, we independently sample simulator parameters $\phi = (g_z, w_x, w_y)$ from the predefined parameter ranges. The kinematic reference is tracked in the sampled environment by the publicly-released MetaMotivo policy (Tirinzoni et al., 2025), and the resulting successful trajectories form the simulated dataset $\mathcal{D}_{\text{sim}}$.

**Training.** We start from a publicly available MDM checkpoint (Tevet et al., 2023), attach the Sim Encoder and Motion Adapters, and train only these newly introduced components. Following MDM's training strategy, we randomly mask the text embedding with probability 10% to enable classifier-free guidance at inference. In contrast, we never mask the simulator embedding $\phi_{\text{sim}}$, as masking these parameters would effectively force the adapters to ignore their inputs. (If physics-free ablations are required, the adapter can simply be disabled by setting the scaling factor $\alpha = 0$.)

The training objective is to minimize the difference between the predicted noise and the true noise using the loss function

$$\mathcal{L} = \mathbb{E}_{\hat{\boldsymbol{\tau}}_0, t, \boldsymbol{c}_{\text{text}}, \boldsymbol{\phi}_{\text{sim}}, \boldsymbol{\epsilon}} \left[ \|\boldsymbol{\epsilon} - \boldsymbol{\epsilon}_\theta(\hat{\boldsymbol{\tau}}_t, t, \boldsymbol{c}_{\text{text}}, \boldsymbol{\phi}_{\text{sim}})\|_2^2 \right]. \tag{11}$$

**Inference.** At inference, we use classifier-free guidance (Ho & Salimans, 2022) to sample motions consistent with both textual prompts and environment parameters. We extend the sampling formulation of the original MDM (Tevet et al., 2023). At each diffusion step $t$, the final prediction is computed as:

$$\tilde{\boldsymbol{\epsilon}}_\theta\big(\boldsymbol{\tau}_t, \boldsymbol{\phi}_{\text{sim}}, \boldsymbol{c}_{\text{text}}\big) = \boldsymbol{\epsilon}_\theta\big(\boldsymbol{\tau}_t, \boldsymbol{\phi}_{\text{sim}}, \varnothing\big) + s_{\text{cfg}} \left(\boldsymbol{\epsilon}_\theta\big(\boldsymbol{\tau}_t, \boldsymbol{\phi}_{\text{sim}}, \boldsymbol{c}_{\text{text}}\big) - \boldsymbol{\epsilon}_\theta\big(\boldsymbol{\tau}_t, \boldsymbol{\phi}_{\text{sim}}, \varnothing\big)\right), \tag{12}$$

where the Motion Adapter remains active in both conditional and unconditional passes, ensuring continuous conditioning on the environment parameters $\phi_{\text{sim}}$. The guidance scale $s_{\text{cfg}}$ is set to 2.5 in all experiments unless stated otherwise.

## 5 EXPERIMENTS

We evaluate whether SimDiff can (i) internalise basic physics constraints to produce physically plausible motions, and (ii) compositionally generalise to previously unseen combinations of environmental conditions.

- **Binary Physical Plausibility:** Can SimDiff generate motions reproducible by a physical tracking controller in a fixed, standard environment (Earth gravity, no wind), without compromising realism and textual alignment?

- **Generalisation to Diverse Environments:** Given explicit environmental parameters $(g, \mathbf{w})$ at inference, can SimDiff successfully adapt to arbitrary combinations of gravity and wind conditions unseen during training?

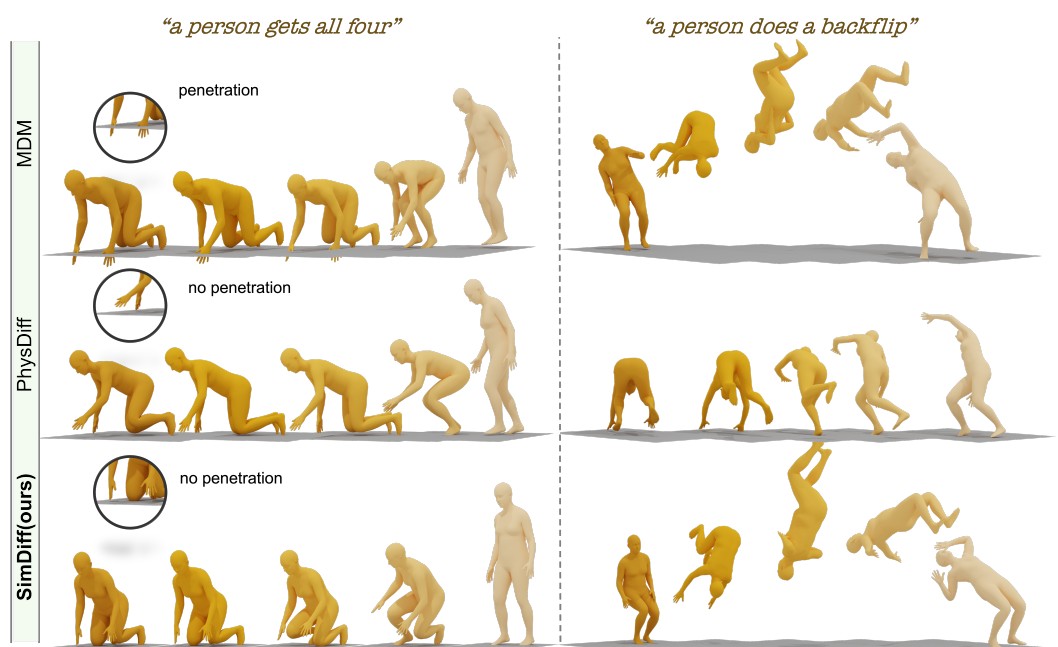

Figure 3: **Visual comparison of MDM, PhysDiff, and SimDiff.** SimDiff preserves balance and clean contacts, whereas MDM shows ground penetration and PhysDiff suffers from instability and tracking errors.

## 5.1 DATASETS

Our experiments are based on the HumanML3D dataset (Guo et al., 2022), a large-scale collection of textually annotated human motions derived from AMASS (Mahmood et al., 2019) and Human-Act12 (Guo et al., 2020). Each HumanML3D sequence provides root-relative joint positions and rotations for 22 body joints. To adapt these motions for simulation, we convert each HumanML3D sequence into SMPL joint rotations using SMPLify (Bogo et al., 2016), running optimization for 100 iterations per sequence. The resulting SMPL representation is then converted into MuJoCo-compatible states by computing root translations, orientations, and joint velocities suitable for physics-based tracking.

## 5.2 EVALUATION METRICS

We adopt standard evaluation metrics from the HumanML3D benchmark (Guo et al., 2020) to comprehensively assess our generated motions from both textual alignment and realism perspectives. For text-to-motion evaluation, we use the *R-Precision*, defined as the accuracy at retrieving the correct text prompt from among 31 randomly sampled negative examples based on a contrastive latent embedding. We quantify realism using the *Frechét Inception Distance (FID)*, measuring the distributional similarity between generated motions and real reference motions, where a lower score indicates higher fidelity. *Multimodal Distance* evaluates the semantic coherence between generated motions and their conditioning texts by computing the mean L2 distance in a learned latent embedding. Finally, *Diversity* is assessed by calculating the variance across generated motions to reflect the model's capacity to produce varied and distinct outputs.

To specifically measure the physical plausibility of generated motions, we also incorporate physics-based metrics proposed in PhysDiff (Yuan et al., 2023). *Penetration* measures the average vertical distance below the ground of any joint that penetrates the floor plane. *Floating* quantifies the average distance above the ground for joints that incorrectly float above the surface, considering a tolerance threshold of 5 mm to account for geometric approximations. Lastly, *Sliding* captures undesirable horizontal sliding movements by averaging horizontal displacements between consecutive frames where ground-contact joints remain within 5 mm of the ground plane. All physics metrics are computed on skeletal joints rather than mesh vertices, following the bone-based protocol used in CloSD (Tevet et al., 2025).

Table 1: Quantitative results for the text-to-motion task on HumanML3D. We highlight in **bold** the better value between PhysDiff and SimDiff (excluding the original MDM baseline) for each metric.

| Method | R Precision↑ Top 3 | Multimodal Dist↓ | FID↓ | Diversity→ | Penetration↓ [mm] | Floating→ [mm] | Sliding↓ [mm] |
|---|---|---|---|---|---|---|---|
| ground truth | 0.746 | 2.95 | 0.001 | 9.51 | 0.000 | 22.796 | 0.206 |
| MDM w/ DDPM | 0.7113 | 3.6446 | 0.4188 | 9.4421 | 0.0463 | 33.5369 | 0.4290 |
| MDM w/ DDIM | 0.7222 | 3.4657 | 0.5806 | 9.8482 | 0.0372 | 30.9683 | 0.4053 |
| PhysDiff w/ DDPM | 0.5994 | 4.3652 | 1.9115 | 8.3069 | 0.0074 | 16.3494 | 0.0145 |
| PhysDiff w/ DDIM | 0.5678 | 4.5681 | 3.4114 | 8.1195 | **0.0009** | 15.2162 | **0.0082** |
| SimDiff w/ DDPM (ours) | 0.7222 | 3.5398 | **0.6473** | **10.0140** | 0.0092 | 19.5425 | 0.1567 |
| SimDiff w/ DDIM (ours) | **0.7386** | **3.4220** | 0.7398 | 10.0234 | 0.0138 | **22.2577** | 0.1697 |

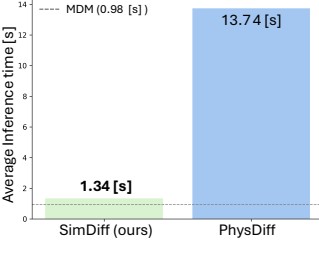

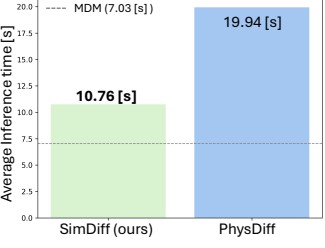

| (a) DDIM (50 steps) | (b) DDPM (1000 steps) |
|---|---|

Figure 4: Inference-speed comparison (batch size = 1) measured on a single NVIDIA A100 GPU.

### 5.3 BINARY PHYSICAL PLAUSIBILITY

**Experimental Setup.** We ask whether SimDiff can embed basic physics when the test environment is kept fixed. Throughout this section the simulator parameters are frozen to $\phi_{\text{sim}} = (g, \mathbf{w})$ with standard gravity $g = -9.8 \text{ m/s}^2$ and no wind $\mathbf{w} = \mathbf{0}$. Every HumanML3D clip is first converted to SMPL (Sec. 5.1) and tracked once by the publicly–released MetaMotivo controller (Tirinzoni et al., 2025). Roll-outs that end in a fall or whose final pose drifts substantially from the reference are discarded; the remaining $18,201$ motions are used to fine-tune SimDiff.

SimDiff starts from the official MDM checkpoint (Tevet et al., 2023) and is trained for 16 epochs ($341,280$ iterations) on $4\times$ NVIDIA A100, batch 64/GPU, Adam ($1\times10^{-4}$) (Kingma & Ba, 2017). At inference we evaluate two samplers. The DDPM sampler uses the full ancestral chain with $1,000$ diffusion steps, exactly as in the official MDM release (Tevet et al., 2023). The DDIM sampler employs a faster, 50-step schedule with the 15–15–8–6–6 respacing proposed in (Song et al., 2020). To assess SimDiff's ability to embed physics plausibility, we compare it against two representative baselines. MDM (Tevet et al., 2023) is the unmodified publicly released model trained only on the original HumanML3D data. PhysDiff (Yuan et al., 2023) uses the recommended "*End4/Space1*" schedule for the 50-step DDIM sampler, while for the 1000-step DDPM sampler it projects at diffusion steps $[60, 40, 20, 0]$.

**Results.** Table 1 summarizes the quantitative comparison between SimDiff (scale $\alpha = 0.1$) and the baseline methods on HumanML3D. First, SimDiff achieves significantly better physics plausibility compared to the original MDM model, substantially reducing penetration (up to $\approx 5\times$ improvement), floating, and sliding artifacts across both DDPM and DDIM samplers. Compared to PhysDiff, SimDiff attains competitive physics metrics—only slightly higher floating and sliding but comparable penetration—while significantly outperforming PhysDiff in motion realism and textual alignment. Specifically, SimDiff shows notably improved *FID*, *R-Precision*, *Multimodal Distance*, and *Diversity*, clearly demonstrating that SimDiff successfully internalizes physics constraints without compromising the original model's generative performance.

These observations are visually confirmed in Figure 3. SimDiff removes penetration artifacts visible in MDM outputs (left side). Furthermore, while PhysDiff often fails to accurately track intended motions due to instability or tracking errors (right side), SimDiff robustly generates the desired motions without compromising realism.

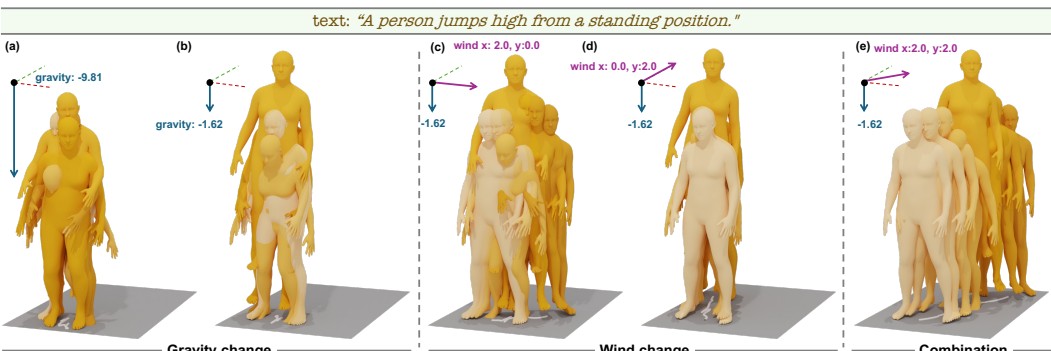

Figure 5: **SimDiff generalises compositionally across gravity and wind conditions.** Left (a–b): varying gravity with no wind; Middle (c–d): introducing wind along X or Y directions; Right (e): combining gravity and diagonal wind—an unseen scenario during training.

Figure 4 compares inference-time [1]. Because SimDiff eliminates the repeated simulator calls required by PhysDiff, it is an order of magnitude faster under the 50-step DDIM sampler and nearly twice as fast under the 1000-step DDPM sampler. These results confirm that SimDiff offers a substantially better speed–quality trade-off, making it practical for real-time or interactive applications.

## 5.4 GENERALISATION TO DIVERSE ENVIRONMENTS

**Experimental Setup.** To evaluate SimDiff's ability to generalise across diverse environments, we generate a total of 30 distinct physics conditions, varying gravity and wind parameters independently. Specifically, gravity conditions are sampled uniformly as $g_z \sim \mathcal{U}[-20, -1]$ m/s$^2$ with no wind ($\mathbf{w} = (0, 0)$), while horizontal wind conditions ($w_x$ or $w_y$) are sampled uniformly from $\mathcal{U}[-10, 10]$ N with gravity fixed at Earth standard ($g_z = -9.81$ m/s$^2$ ). Only one environmental parameter is changed at a time during dataset creation. We replay all motions from HumanML3D using the Meta-Motivo controller in these environments, discarding motions that cannot be accurately tracked. The successfully tracked motions form our simulator-augmented dataset for training. SimDiff is trained on this data for $129, 130$ iterations with a batch size of $128$ and a learning rate of $1 \times 10^{-4}$ (Adam optimiser (Kingma & Ba, 2017)). Only the Sim Encoder and Motion Adapters are trained.

**Results.** Figure 5 illustrates SimDiff's ability to generalise compositionally across gravity and wind conditions. In the gravity-varying cases (a–b), reducing gravity clearly leads to increased jump heights, matching physical intuition. Cases (c–d) demonstrate that SimDiff accurately conditions motions on horizontal wind, causing trajectories to drift consistently in the wind's direction (see motion traces on the ground) while maintaining the higher jump achieved under reduced gravity. In the combined gravity-and-diagonal-wind case (e), SimDiff simultaneously respects wind conditions in both X and Y directions, resulting in pronounced diagonal displacement along the wind axes without compromising jump height. These results demonstrate that SimDiff successfully generalises beyond the training conditions, in which only one environmental parameter was varied at a time.

## 6 CONCLUSION

We presented **SimDiff**, a simulator-constrained diffusion model that directly integrates physical constraints into the denoising process by explicitly conditioning on environmental parameters. By training on motion data generated across a variety of physical conditions, SimDiff successfully synthesises physically plausible motions without requiring expensive simulator-based corrections at inference, and robustly generalises to unseen multi-factor scenarios. Additionally, our reformulation of simulator-based motion projection as classifier guidance provides insights into how external physics simulators can effectively steer diffusion models. Future work includes extending SimDiff to handle richer environmental interactions, such as uneven terrain, as well as conditioning on additional character-specific parameters, such as joint angles and body morphology.

---

[1]Note that, for fairness, PhysDiff's inference times here exclude the additional runtime overhead from inverse-kinematics (IK) steps and include only the simulator projections.

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

## A    ADDITIONAL RESULTS AND VIDEOS

Additional qualitative results, including video comparisons, are available on our supplementary website:

```
https://simdiff-anonymous.github.io/simdiff_anonymous.github.io/
```

The supplementary results are organized into three categories:

**Benchmark Prompts.** We visually compare motions generated by original MDM (Tevet et al., 2023), PhysDiff (Yuan et al., 2023), and our proposed SimDiff across representative HumanML3D prompts (*adjusting sitting position*, *backflip*, and *crawling*). SimDiff generates physically plausible motions while preserving stylistic and semantic details.

**Single-Parameter Variations.** To demonstrate direct environment control, we independently vary gravity and planar wind parameters. Annotated sliders indicate the intensity of these parameters. Changes in gravity clearly affect airtime and vertical displacement, while planar wind causes horizontal shifts in the wind direction.

**Compositional Generalization.** We present motions under novel environmental combinations (low gravity with diagonal wind) unseen during training. Results illustrate that SimDiff successfully generalizes by producing physically plausible motions responsive to multiple simultaneous environmental changes.

All motions were visualized using the SMPL mesh (Loper et al., 2015), optimized with SMPLify (Bogo et al., 2016) for 2,000 iterations using the L-BFGS optimizer (Liu & Nocedal, 1989) , and rendered in Blender. For consistent viewing, we fixed the random seed and kept the camera height (along the $z$-axis) constant for all clips of the same prompt.

## B    SIMULATION ENVIRONMENT

All physics roll-outs are executed using the MetaMotivo (Tirinzoni et al., 2025) environment built on the MuJoCo (Todorov et al., 2012) physics simulator. We specifically employ the largest publicly released model, `metamotivo-M-1` (228M parameters).

## C    PHYSDIFF REIMPLEMENTATION DETAILS

For a fair comparison, we re-implemented the PhysDiff projection module (Yuan et al., 2023) within the same MetaMotivo environment used for SimDiff and matched every setting to those employed during SimDiff data generation. The original PhysDiff relies on a Residual Force term, an auxiliary external force field used to compensate for dynamics mismatch (Yuan & Kitani, 2020). We disable this Residual Force so that the character moves under purely internal torques, making our setup closer to realistic, force-free motion.

## D    DATASET FILTERING PROTOCOL

Prior to training, we applied a filtering step to the tracked HumanML3D dataset to exclude motions whose physics-tracked rollouts significantly diverged from their original kinematic reference motions. Both the original reference sequences $\tau^{\mathrm{ref}}$ and the tracked sequences $\tau^{\mathrm{trk}}$ were represented in the HumanML3D format, where each frame encodes joint positions relative to the preceding frame and root orientations in the local character coordinate frame.

To quantify the divergence between a tracked rollout and its reference, we computed the mean positional discrepancy across all joints and frames in the HumanML3D representation:

$$d_{L_2}\left(\tau^{\mathrm{ref}}, \tau^{\mathrm{trk}}\right) = \frac{1}{N} \sum_{n=1}^{N} \left\|\tau_n^{\mathrm{ref}} - \tau_n^{\mathrm{trk}}\right\|_2. \tag{13}$$

A motion pair $(\tau^{\mathrm{ref}}, \tau^{\mathrm{trk}})$ was retained if its mean positional discrepancy satisfied:

$$d_{L_2}\left(\tau^{\mathrm{ref}}, \tau^{\mathrm{trk}}\right) \leq \tau_{L_2}, \tag{14}$$

Table 2: Quantitative results for SimDiff with DDIM on HumanML3D under different adapter scales.

| Adapter Scale $\alpha$ | R Precision↑ | Multimodal | FID↓ | Diversity→ | Penetration↓ [mm] | Floating→ [mm] | Sliding↓ [mm] |
|---|---|---|---|---|---|---|---|
| | Top-3 | | | | | | |
| 0.01 | 0.7289 | 3.4511 | 0.5867 | 9.9143 | 0.0291 | 29.5284 | 0.3670 |
| 0.05 | 0.7358 | 3.4302 | 0.6493 | 9.9733 | 0.0180 | 25.9204 | 0.2609 |
| 0.10 | 0.7386 | 3.4220 | 0.7398 | 10.0234 | 0.0138 | 22.2577 | 0.1697 |
| 0.20 | 0.7369 | 3.4262 | 0.9197 | 9.9931 | 0.0102 | 16.6901 | 0.0835 |
| 0.30 | 0.7386 | 3.4204 | 1.0313 | 9.9537 | 0.0164 | 13.4422 | 0.0400 |
| 0.40 | 0.7403 | 3.4110 | 1.1164 | 9.8795 | 0.0180 | 12.0631 | 0.0216 |
| 0.50 | 0.7386 | 3.4030 | 1.1896 | 9.7724 | 0.0132 | 11.7828 | 0.0175 |
| 0.60 | 0.7369 | 3.4372 | 1.2693 | 9.6999 | 0.0125 | 12.0161 | 0.0126 |
| 0.70 | 0.7341 | 3.4781 | 1.3627 | 9.5699 | 0.0068 | 12.5174 | 0.0110 |
| 0.80 | 0.7205 | 3.5378 | 1.4878 | 9.3586 | 0.0036 | 13.1213 | 0.0118 |
| 0.90 | 0.7089 | 3.6088 | 1.6542 | 9.1211 | 0.0041 | 13.7288 | 0.0143 |
| 1.00 | 0.6926 | 3.7348 | 1.9055 | 8.8614 | 0.0081 | 14.5198 | 0.0154 |

with the threshold $\tau_{L_2} = 7.0$ selected empirically by visually inspecting representative examples. This threshold preserved motions that closely followed their original semantics while excluding obvious tracking failures, such as unrealistic drifting or falling motions.

# E  ARCHITECTURE DETAILS

## E.1  SIM ENCODER

The Sim Encoder processes environmental parameters into a 512-dimensional embedding compatible with the Transformer hidden states. We consider two configurations based on the environmental inputs:

- *Categorical encoding (tracked motions only):* We encode environment information categorically using only one active class (tracked conditions). This is implemented by embedding a single categorical index into a 64-dimensional vector, followed by a linear projection to 512 dimensions.

- *Continuous parameters:* Environment parameters $(g_z, w_x, w_y)$ are directly projected from 3-dimensional continuous inputs to 64 dimensions via a linear layer, followed by another linear projection to 512 dimensions.

## E.2  MOTION ADAPTER

Each Motion Adapter employs a bottleneck structure with the following dimensions:

- Input dimension: 512
- Bottleneck dimension: 256
- Environment feature dimension: 512

Two Motion Adapters are integrated into each of the 8 Transformer layers—one after the self-attention module and one following the feed-forward network—yielding 16 adapters in total. The up-projection layers within each adapter are initialized with zeros to ensure stable adaptation during the early stages of fine-tuning.

# F  ABLATION STUDY ON ADAPTER SCALE

Table 2 reports quantitative results of SimDiff using the DDIM sampler under varying adapter scales $\alpha$. All evaluations are conducted on HumanML3D. Smaller adapter scales yield better perceptual quality (lower FID), but show higher physical artifacts such as floating and sliding. As $\alpha$ increases, physics-related errors significantly decrease.

