# OpenReview forum: "SimDiff: Simulator-constrained Diffusion Model for Physically Plausible Motion Generation"
_ICLR.cc/2026/Conference — Submitted to ICLR 2026_

### Official Review · Reviewer_vsFh · 2025-10-28

**Soundness:** 3
**Presentation:** 4
**Contribution:** 3
**Rating:** 4
**Confidence:** 4

**Summary:**

The paper presents a method for motion generation that is more physically accurate than traditional diffusion-based generation. The main idea is using guidance to push the generation towards more plausible generations. While this idea is sound, efficient, and easy to implement. It is not very surprising, and its application is rather niche. For example, the paper states that as opposed to other approaches, the guidance-based approach is more computationally efficient, however motion generation models are rather slim, and a use-case where this add cost is crucial for is not presented.
Perhaps most importantly, the presented results and evaluation are accordingly underwhelming. Against a rather dated method, some improvement in quality can be observed in the handful of demonstrated qualitative results. However even these selected motions are still not physically plausible, with minimal improvement in quality, and seem to be less prompt adherent. The quantitative evaluation supports this overall impression as well.

**Strengths:**

The paper is concise and to the point, with decent theoretical analysis, and a simple to follow. The method is simple to implement and understand. Some quality improvements can be observed.

**Weaknesses:**

- The paper presents an unclear usecase - of computationally efficient MDM-based generation with more physical plausibility: Why is the computational efficiency important? Why use a dated version of MDM when newer autoregressive versions of MDM (e.g., CAMDM, CLosD) have shown to have better performance?
- The paper compares to Physdiff, which is a rather dated method. There have been many attempts to incorporate physics into human motion generation, including UniHSI, CHOIS, CLoSD (which is mentioned by the authors), and more. All these papers also show more applications to physical simulation, like object interaction. Does this method apply to such cases as well?
- The qualitative evaluation is slim and unpersuasive
- The quantitative evaluation is also slim and unpersuasive: comparing only to one previous and dated method is not enough, and the numbers show worse penetration (which is one of the only aspects presented as a usecase), floating that is similar to the GT (which is not great actually, it should have been improved), and FID which is comparable to the original MDM (which is good, meaning minimal harm) but not as good as state-of-the-art motion generation.

**Questions:**

See questions in weaknesses.
Overall the idea is sound, but does not pass the bar for a prestigious venue such as ICLR in this reviewer's opinion.

---

> ### Author Response · Authors · 2025-11-23
> **Official Comment by Authors (Part 1/2)**
>
> We appreciate the valuable and thoughtful feedback from the reviewer. We respond to the concrete questions and comments in detail below.
>
> > The paper presents an unclear usecase - of computationally efficient MDM-based generation with more physical plausibility: Why is the computational efficiency important? Why use a dated version of MDM when newer autoregressive versions of MDM (e.g., CAMDM, CLosD) have shown to have better performance?
>
> We appreciate the question and clarify both the use case and backbone choice. First, the importance of computational efficiency is paramount. A rapidly expanding area of research is dedicated to real-time applications, such as feeding generative models into low-level controllers for robotics, as exemplified by recent approaches like CloSD[1]. In this context, inference speed is the critical bottleneck. Our method's 10x speedup over a strong baseline (as shown in Fig. 4) is precisely its core advantage, enabling practical use in these interactive scenarios.
> Second, we would like to clarify the choice of our backbone. This choice does not detract from our core contribution. Models like CLoSD, which the reviewer terms "autoregressive," are not so in the typical sense. Their fundamental architecture is still based on MDM[2]: they generate full sequences conditioned on prior segments from the simulator. Therefore, our contribution, lightweight adapters for physics conditioning, is orthogonal and can be applied directly to CLoSD and any other MDM-based model. Our proof on the foundational MDM is sufficient, as the conclusions will hold regardless of which MDM-based architecture is used.
>
>
> > The paper compares to Physdiff, which is a rather dated method. There have been many attempts to incorporate physics into human motion generation, including UniHSI, CHOIS, CLoSD (which is mentioned by the authors), and more. All these papers also show more applications to physical simulation, like object interaction. Does this method apply to such cases as well?
>
> Yes, it is absolutely applicable to those cases. But we respectfully disagree with the reviewer’s characterization of these methods and their relationship to our work. UniHSI, CHOIS, and CLoSD operate at fundamentally different layers of the motion-generation stack. Understanding this distinction is important for determining whether SimDiff “applies” to those settings.
>
> - UniHSI[3] focuses on low-level control: it executes plans generated by an LLM and does not use diffusion models at all, so it is not a baseline for high-level motion generation.
>
> - CHOIS[4] addresses a different task because it jointly synthesizes object trajectories, whereas SimDiff generates only human motion; our model could supply the human-motion component inside CHOIS, but CHOIS is not a text-to-motion baseline.
>
> - CLoSD[1] is the closest conceptually, since it combines a diffusion planner with a low-level controller, but its evaluation focuses on controller success rates in closed-loop tasks. Notably, CLoSD itself does not compare against PhysDiff, even though PhysDiff could serve as the high-level planner, simply because PhysDiff is far too slow for their closed-loop setting, not because it is “dated.” Our contribution addresses exactly this bottleneck by providing a fast, simulator-free planner that could be plugged into such systems if desired.
>
> Overall, SimDiff is best viewed as a generic, environment-aware text-to-motion planner that complements UniHSI, CHOIS, and CLoSD rather than competing with their specific control and interaction setups. For this reason we compare primarily to MDM and PhysDiff, which share our text-to-motion setting, and leave applying SimDiff within UniHSI/CHOIS/CLoSD-style stacks and interaction benchmarks as an important but orthogonal direction for future work.

---

> ### Author Response · Authors · 2025-11-23
> **Official Comment by Authors (Part 2/2)**
>
> > The qualitative evaluation is slim and unpersuasive. The quantitative evaluation is also slim and unpersuasive: comparing only to one previous and dated method is not enough, and the numbers show worse penetration (which is one of the only aspects presented as a usecase), floating that is similar to the GT (which is not great actually, it should have been improved), and FID which is comparable to the original MDM (which is good, meaning minimal harm) but not as good as state-of-the-art motion generation.
>
> We would like to clarify our rationale for the baseline comparison, as this seems to be a point of misunderstanding. When evaluating physical plausibility, the simulator itself functions as the oracle because it is the definitive source of ground-truth physics. For this reason, PhysDiff, which directly calls the simulator at inference time to project each denoised sample, is not a “dated” or weak baseline. Its design explicitly optimizes for the same physics-based quantities measured in our evaluation. We also note that other recent methods, such as ReinDiffuse[5], did not report a direct quantitative comparison of physics metrics against PhysDiff, which indicates that PhysDiff remains a challenging reference point. In this light, benchmarking against PhysDiff serves not as a comparison to a dated method, but as a rigorous evaluation against a strong baseline that explicitly enforces physical correctness through simulation. On the specific metrics the reviewer cites, SimDiff intentionally trades some of PhysDiff’s per‑step projection for speed and text fidelity: relative to MDM it reduces penetration from 0.0372 mm to 0.0138 mm and sliding from 0.4053 mm to 0.1697 mm while achieving much lower FID than PhysDiff. Table 2 in the Appendix further shows that increasing the adapter scale reduces penetration, floating, and sliding if one prefers stronger physics at some FID cost, which makes the tradeoff explicit (Table 1 and Table 2).
>
>
>
> ---
>
> [1] Tevet G, Raab S, Cohan S, et al. CLoSD: Closing the Loop between Simulation and Diffusion for multi-task character control[C]//The Thirteenth International Conference on Learning Representations (ICLR). 2025.
>
> [2] Tevet G, Raab S, Gordon B, et al. Human Motion Diffusion Model[C]//The Eleventh International Conference on Learning Representations (ICLR). 2023.
>
> [3] Xiao Z, Wang T, Wang J, et al. Unified Human-Scene Interaction via Prompted Chain-of-Contacts[C]//The Twelfth International Conference on Learning Representations (ICLR). 2024.
>
> [4] Li J, Clegg A, Mottaghi R, et al. Controllable Human-Object Interaction Synthesis[C]//Computer Vision – ECCV 2024, Lecture Notes in Computer Science, vol. 15099. Springer, 2025: 54–72.
>
> [5] Han G, Liang M, Tang J, et al. ReinDiffuse: Crafting Physically Plausible Motions with Reinforced Diffusion Model[C]//2025 IEEE/CVF Winter Conference on Applications of Computer Vision (WACV). IEEE, 2025: 2218–2227.

---

### Official Review · Reviewer_ZHbu · 2025-10-30

**Soundness:** 2
**Presentation:** 2
**Contribution:** 3
**Rating:** 4
**Confidence:** 3

**Summary:**

This paper rethinks simulator-based motion projection (as used in previous papers such as PhysDiff) as guided diffusion. The authors propose SimDiff, which conditions a text-to-motion diffusion model on environment parameters (specifically gravity and wind). Instead of invoking a simulator at every step to generate physically plausible motion, SimDiff trains on simulator-augmented guidance. SimDiff uses classifier-free guidance with the parameters to steer the samples towards physically plausible motions. Experiments report better physics metrics than MDM and faster sampling than a modified PhysDiff baseline, along with qualitative compositional generalization.

**Strengths:**

- Amortizing physics using guided diffusion replaces the need to call a simulator at each reverse step, leading to big computational gains.
- Conceptual bridge between projection and guidance.
- Experiments show faster sampling than sim-in-the-loop PhysDiff at similar step counts, and demonstrate initial environmental control.
- Conversion of HumanML3D to SMPL using SMPLify. Overall solid data pipeline.
- Clear architectural choice of training motion adapters

**Weaknesses:**

- **Weak related work**: Section 2.1, the related work on human motion generation using diffusion, is not nearly expansive enough. It only cites 3 previous works on human motion generation with diffusion despite the literature that has come out in this area. As a result, the positioning of this paper contextually is not strong. Here are some more papers which can be cited.
  - Goal-Driven Human Motion Synthesis in Diverse Tasks, CVPR 2025 HuMoGen.
  - HOI-Diff: Text-Driven Synthesis of 3D Human-Object Interactions using Diffusion Models, CVPR 2025 HuMoGen.
  - Fitness-Aware Human Motion Generation with Fine-Tuning, NeurIPS FITML 2024
  - Constrained Synthesis with Projected Diffusion Models, Christopher et al, NeurIPS 2024
  - Aligning Human Motion Generation with Human Perceptions, Wang et al, ICLR 2025
- **Narrow conditioning scope in experiments**: Section 4.2 motivates conditioning on gravity and friction as environmental parameters, but only gravity and planar wind are varied in the experiments. Conditioning is a small set of global scalars $[g_z, w_x, w_y]$.
- **Claim on generalization to unseen parameter combinations**: The abstract claims SimDiff "successfully generalizes to unseen combinations of environmental parameters, demonstrating compositional generalization." However, the evidence is qualitative only with no quantitative compositional metric, and the conditioning space is only limited to global scalars in gravity and planar wind. This makes the claim **narrowly validated** - it shows qualitative composition of two global coefficients, not transfer to spatially varying or contact-rich regions. Without quantitative metrics or broader parameter types, this claim should be read as modest in scope.
- **Selection bias in $\mathcal{D}_\text{sim}$**: In sections 5.3/5.4 and Appendix D, the authors state that **failed** controller rollouts are discarded when constructing the training set. This biases the learned conditional towards what *this particular* controller and contact model can track, rather than the full set of physically plausible motions. This risks degradation under parameter mismatch or different controllers. The controller's stabilization choices define what counts as success. PhysDiff adds a Residual Force term to compensate dynamics mismatch, which SimDiff's reproduction disables.
- **Comparison to PhysDiff**: Appendix C states that the Residual Force term in PhysDiff is disabled, which is used to stabilize hard behaviors. This can understate PhysDiff's stability.
- **Limited physics evaluation**: metrics are joint-based (not mesh-based), with no contact-force/torque statistics and no closed-loop tracking success under a different controller.

**Questions:**

- How do text fidelity and physics metrics degrade when gravity/wind are misspecified at sampling time? It would be good to include curves vs $| \Delta g|$ and wind-angle error.
- Why did you remove the residual force term in PhysDiff? Can you report PhysDiff **with** RF in the MuJoCo port?
- Do you have concrete projection schedules + wall-clock time?
- Have you tested a subset of the experiments in section 5.3 using a different controller to see how the method reacts to an alternative controller?
- Figure 5 shows qualitative composition; can compositional generalization be quantified?
- Is classifier-free guidance essential, or could similar effects be obtained with learned conditioning through adapters alone?

---

> ### Author Response · Authors · 2025-11-23
> **Official Comment by Authors (Part 1/3)**
>
> We appreciate the valuable and thoughtful feedback from the reviewer. We respond to the concrete questions and comments in detail below.
>
> > Why did you remove the residual force term in PhysDiff? Can you report PhysDiff with RF in the MuJoCo port?
>
> We did not intentionally remove the residual force. In the MetaMotivo setup, the residual force is off by default, and our MuJoCo port simply kept that default. As stated in the MetaMotivo paper [1], their controller uses “a proportional derivative (PD) controller and the action space (A \subseteq [-1,1]^{69}) thus specifies the ‘normalized’ PD target. Unlike previous work, which considered an under-constrained skeleton and over-actuated controllers, we define joint ranges and torque limits to create ‘physically plausible’ movements.” This description makes clear that MetaMotivo does not rely on auxiliary external forces. Turning residual force on is not a toggle in this setting because doing so would require expanding the action space and retraining the low-level policy. Modern controllers such as MaskedMimic [2] and Perpetual Humanoid Control [3] are likewise designed to operate without the residual force. For this reason, evaluating with residual force off is the appropriate choice when using a modern imitation controller.
>
> > Do you have concrete projection schedules + wall-clock time?
>
> Yes, both the projection schedules of PhysDiff and the wall-clock timings are already provided in the paper. The detailed inference-time comparison appears in Figure 4 on page 8, lines 374-377, where we report the exact wall-clock times measured on an NVIDIA A100. With 50 DDIM steps, SimDiff requires 1.34 s, while PhysDiff requires 13.74 s under the same settings.  The projection schedules for PhysDiff are described in Section 5.3, page 7, lines 349-352, where we specify that PhysDiff uses the “End4/Space1” schedule for DDIM following the original PhysDiff paper and performs projections at diffusion steps [60, 40, 20, 0] for DDPM.
>
> > Have you tested a subset of the experiments in section 5.3 using a different controller to see how the method reacts to an alternative controller?
>
> No. We did not run Section 5.3 with a different controller, and believe that doing so would not materially change the relative conclusion we report. Both SimDiff and PhysDiff depend on the same physics projection operator $P_{\phi,\pi}$: SimDiff uses $P_{\phi,\pi}$ to build the simulator‑augmented training corpus, and PhysDiff invokes the same operator during its inference‑time projections. Changing the controller would change $P_{\phi,\pi}$ for both methods in the same direction, which conflates planner and controller and does not isolate the planners. We therefore kept $P_{\phi,\pi}$ fixed across methods.

---

> ### Author Response · Authors · 2025-11-23
> **Official Comment by Authors (Part 2/3)**
>
> > Figure 5 shows qualitative composition; can compositional generalization be quantified?
>
> Thank you for this valuable suggestion! We acknowledge that quantifying compositional generalization in high-dimensional human motion is a challenging task, often limited to simple qualitative visualizations in prior work[4,5]. However, to provide a more rigorous assessment as requested, we developed a quantitative metric based on the linearity of physical effects.
>
> Compositional generalization here refers to SimDiff's capability to accurately predict the combined effect of multiple physical conditions from motions trained on each condition independently. Consider the scenario shown in Figure 5, where lunar gravity and horizontal wind conditions are combined. Let $\Delta x$ denote the pelvis displacement under horizontal wind blowing in the x-direction alone (with lunar gravity and no y-direction wind), and $\Delta y$ denote the pelvis displacement under horizontal wind blowing in the y-direction alone (with lunar gravity and no x-direction wind). Then, $\Delta xy$ is the pelvis displacement under simultaneous horizontal winds in both the x- and y-directions. If SimDiff generalizes compositionally, we expect $\Delta xy$ to be well approximated by a linear combination of the independent effects $\Delta x + \Delta y$. We quantitatively measure this using the coefficient of determination ($R^2$), which indicates how closely the combined displacement ($\Delta xy$) matches this linear approximation ($\Delta x + \Delta y$).
>
> To quantitatively evaluate compositional generalization, we specifically analyzed and quantified the scenario presented in Figure 5. We measured pelvis displacement under different horizontal wind conditions combined with lunar gravity (g = -1.62 m/s²). The official MDM baseline was evaluated using textual descriptions explicitly specifying each condition combination, without separate physics parameters:
> - "a person jumps high from a standing position under lunar gravity (g=-1.62 m/s²) with no wind"
> - "a person jumps high from a standing position under lunar gravity (g=-1.62 m/s²) with horizontal wind wx=2.0 m/s² and wy=0.0 m/s²"
> - "a person jumps high from a standing position under lunar gravity (g=-1.62 m/s²) with horizontal wind wx=0.0 m/s² and wy=2.0 m/s²"
> - "a person jumps high from a standing position under lunar gravity (g=-1.62 m/s²) with diagonal wind wx=2.0 m/s² and wy=2.0 m/s²"
> The resulting $R^2$ values, which measure how closely the combined horizontal displacement ($\Delta xy$) matches the linear approximation ($\Delta x + \Delta y$), are summarized below:
> | Method         | $R^2$ (higher is better) |
> | -------------- | --------------------- |
> | MDM | 0.224                 |
> | SimDiff (ours) | **0.870**             |
>
> The substantially higher $R^2$ achieved by SimDiff quantitatively demonstrates our model's superior compositional generalization capability compared to the original MDM model.
>
>
>
>
>
> >Is classifier-free guidance essential, or could similar effects be obtained with learned conditioning through adapters alone?
>
> CFG is not essential for producing physics effects in SimDiff. As explained in the Training and Inference subsection (Section 4.2), we mask the text embedding to enable classifier-free guidance but never mask the simulator embedding. Consequently, the Motion Adapters always receive the environment parameters and inject the physics signal regardless of whether the text branch is active. The sampling formulation defined in Eq. (12) further makes this explicit: the Motion Adapter is active in both the conditional and unconditional passes, so both branches already contain the same environment-conditioned physics information. Adjusting $s_{\text{cfg}}$ therefore only reweights two physics-aware predictions to tune text adherence, rather than enabling or disabling the physical effects. Physics is governed by the adapters themselves, which is why our ablation varies the adapter coefficient $\alpha$. As shown in Appendix Table 2 (page 15), increasing $\alpha$ consistently reduces penetration, floating, and sliding at a modest FID cost, confirming that physics adherence arises from the adapters, not from CFG.

---

> > ### Author Response · Authors · 2025-11-23
> > **Official Comment by Authors (Part 3/3)**
> >
> > [1] Tirinzoni A, Touati A, Farebrother J, et al. Zero-Shot Whole-Body Humanoid Control via Behavioral Foundation Models[J]. arXiv preprint arXiv:2504.11054, 2025.
> >
> > [2] Tessler C, Guo Y, Nabati O, Chechik G, Peng X B. MaskedMimic: Unified Physics-Based Character Control Through Masked Motion Inpainting[J]. ACM Transactions on Graphics, 2024, 43(6): 209.
> >
> > [3] Luo Z, Cao J, Winkler A, Kitani K, Xu W. Perpetual Humanoid Control for Real-time Simulated Avatars[C]//IEEE/CVF International Conference on Computer Vision (ICCV). 2023: 10861–10870.
> >
> > [4] Janner M, Du Y, Tenenbaum J B, Levine S. Planning with Diffusion for Flexible Behavior Synthesis[C]//International Conference on Machine Learning (ICML). 2022.
> >
> > [5] Ajay A, Du Y, Gupta A, Tenenbaum J B, Jaakkola T S, Agrawal P. Is Conditional Generative Modeling all you need for Decision-Making?[C]//International Conference on Learning Representations (ICLR). 2023.

---

### Official Review · Reviewer_gF3b · 2025-10-31

**Soundness:** 2
**Presentation:** 3
**Contribution:** 2
**Rating:** 4
**Confidence:** 4

**Summary:**

The paper proposes SimDiff, a simulator-constrained diffusion model that integrates physical constraints directly into the diffusion process using classifier-free guidance. It is able to generalize to unseen combinations of environmental parameters.

**Strengths:**

1. The idea of using physics guidance in the diffusion step is interesting.

2. Ability to generalize to unseen explicit physical parameters is a plus.

**Weaknesses:**

1. The statement “approaches have not yet provided (i) a principled way to modify the diffusion steps so that environment-specific physics constraints are taken into account” is too absolute and not fully accurate. Several lines of work already modify the denoising trajectory in a principled, per-step manner [1,2,3]. While many of these methods may not explicitly condition on environment parameters (gravity, wind, friction) or provide a calibrated physics likelihood, they do constitute principled step-level mechanisms beyond heuristic scaling.

2. The method uses a physical simulator as the guidance in the diffusion step, but lacks the explicit control or modelling of the physical motions.

3. Why is “modify the diffusion steps so that environment-specific physics constraints are taken into account” important? What benefit does it bring compared with other physics-guided methods, theoretically and empirically? Regarding the compositional generalization, the comparison with baselines is not presented.


4. My major concerns are with the experiments conducted. (1). Only one dataset is evaluated; more commonly used benchmarks, like KIT-ML, HumanAct12, and UESTC, are needed. Otherwise, it is hard to justify the capability of the method on various datasets. (2). The reported scores for PhysDiff are pretty low compared to the results in their paper. For instance, the reported FID of PhysDiff is 0.433, while in your Table 1, it is 3.411. The difference is huge. Please explain, and justify if it is a fair comparison. (3). As the paper aims for a better way of incorporating the physical guidance, it should compare with more baselines in this direction. Current comparison is not enough. (4). Section 5.4, “Generalisation to diverse environments”. Only qualitative results are provided. More comprehensive comparisons, both qualitative and quantitative, are needed.


5. Ablation studies are not sufficient, e.g., the guidance scale, the key components.


[1]. Christopher J K, Baek S, Fioretto N. Constrained synthesis with projected diffusion models[J]. Advances in Neural Information Processing Systems, 2024, 37: 89307-89333.

[2]. Li Z, Luo M, Hou R, et al. Morph: A Motion-free Physics Optimization Framework for Human Motion Generation[C]//Proceedings of the IEEE/CVF International Conference on Computer Vision. 2025: 14580-14589.

[3]. Han G, Liang M, Tang J, et al. Reindiffuse: Crafting physically plausible motions with reinforced diffusion model[C]//2025 IEEE/CVF Winter Conference on Applications of Computer Vision (WACV). IEEE, 2025: 2218-2227.

**Questions:**

See my weakness

---

> ### Author Response · Authors · 2025-11-23
> **Official Comment by Authors (Part 1/5)**
>
> We appreciate the valuable and thoughtful feedback from the reviewer. We respond to the concrete questions and comments in detail below.
>
> >The statement “approaches have not yet provided (i) a principled way to modify the diffusion steps so that environment-specific physics constraints are taken into account” is too absolute and not fully accurate. Several lines of work already modify the denoising trajectory in a principled, per-step manner [1,2,3]. While many of these methods may not explicitly condition on environment parameters (gravity, wind, friction) or provide a calibrated physics likelihood, they do constitute principled step-level mechanisms beyond heuristic scaling.
>
> We thank the reviewer for pointing this out and agree that our original phrasing was too absolute. Prior work, such as Constrained Synthesis with Projected Diffusion Models (PDM) [1], Morph [2], and ReinDiffuse [3] already propose well‑defined ways of modifying the denoising trajectory at each diffusion step, either via explicit projections onto constraint sets, physics‑based optimisation, or reinforcement learning with physics‑aware rewards. We will revise the sentence in Sec. 1 to explicitly acknowledge these works and remove the suggestion that “no principled modification of the diffusion steps” existed as follows:
>
> “(i) a standalone physics‑aware text‑to‑motion generative model that internalises simulator behaviour, eliminating the need for any simulator or controller at test time, …”
>
> >The method uses a physical simulator as the guidance in the diffusion step, but lacks the explicit control or modelling of the physical motions.
>
>
> This comment misses the main design goal of our approach. The objective of our method is to avoid hand‑crafted, explicit physics models: the dynamics of motion are encoded implicitly in a diffusion model trained on physically realized trajectories. In this paper, those trajectories happen to be generated in simulation, but the algorithm itself is agnostic to the data source and could equally be trained on trajectories collected from a real robot with no change. This data‑driven formulation contrasts with explicit modelling approaches, which are tied to a particular hand‑crafted dynamics model or controller and do not directly benefit from additional real‑world trajectory data. In our case, incorporating experience from a new robot simply amounts to collecting its trajectories and retraining the same diffusion model.

---

> ### Author Response · Authors · 2025-11-23
> **Official Comment by Authors (Part 2/5)**
>
> >Why is “modify the diffusion steps so that environment-specific physics constraints are taken into account” important? What benefit does it bring compared with other physics-guided methods, theoretically and empirically?
>
> Physical constraints are often non-linear and non‑convex, so a complete general theory for physics‑guided diffusion is inherently difficult. Even the projected diffusion work that the reviewers cited [1] gives guarantees only in a convex setting and for relatively simple constraint sets. Such approach is still a useful comparison point because it makes explicit what is guaranteed when constraints are enforced by projection inside the sampler.
>
> On the theoretical side, both SimDiff and PDM can be viewed as trying to sample from a posterior of the form $p(\boldsymbol{\tau} \mid \mathcal{Y}=1)$, where $\mathcal{Y}=1$ denotes “the trajectory satisfies the desired physics constraints in the given environment”. SimDiff directly learns the conditional score $\nabla_\boldsymbol{\tau} \log p(\boldsymbol{\tau}_t \mid \mathcal{Y}=1)$. Conditional score‑based theory [4] shows that the conditional denoising estimator is a consistent estimator of the conditional score in infinite‑dimensional inverse problems (Sec. 5, Proposition 4), and that using this conditional score in the reverse SDE drives the sampler toward the conditional target distribution (Sec. 4–5). Thus, although our setting is more complex, it does say that learning a conditional score field is the theoretically principled direction when the goal is to approximate $p(\boldsymbol{\tau} \mid Y=1)$. In contrast, PDM keeps an unconditional score and enforces feasibility by projecting each iterate back to the constraint set. Its analysis proves that the update remains feasible and converges under convex constraints (Sec. 4.1, Theorem 4.2 “Algorithm validity” in [1]), but does not guarantee faithfulness to the original data distribution. The authors themselves remark that “high projection cost results in significant divergence from the distribution in all settings we explored” (see Sec. 7 in [1]).
>
> Empirically, putting physics into the denoiser removes all per‑step constrained optimizations or simulator calls. SimDiff therefore keeps runtime close to a standard text‑to‑motion model while still controlling gravity, wind, and other environment parameters at test time. Projection‑based methods must solve an inner optimization at every step, and when the constraints become more realistic, this projection essentially degenerates into running a simulator. This is exactly why PhysDiff is slow even though its physics are accurate. PDM demonstrates this overhead even for relatively simple constraints  (Appendix F, Table 2 in [1]). ReinDiffuse avoids simulators in the loop but instead relies on a reward model and RL with hand‑designed physics rewards, which must be reshaped for each new set of constraints and does not expose environment parameters as clean conditioning variables. SimDiff needs neither per‑step optimization nor reward shaping. It learns a conditional score once, then generates fast samples that remain physically plausible under different environments by simply changing the conditioning input.
>
> >My major concerns are with the experiments conducted. (1). Only one dataset is evaluated; more commonly used benchmarks, like KIT-ML, HumanAct12, and UESTC, are needed.
>
>
> We agree that broader coverage is valuable, but we intentionally focused our compute on HumanML3D because it is common in recent works, such as CondMDI[5] and DNO[6] that evaluate exclusively on HumanML3D. Regarding HumanAct12, which the reviewer specifically mentioned, it is already included within HumanML3D.  Therefore, reporting HumanAct12 again as a separate benchmark would amount to measuring the same underlying motions twice.

---

> ### Author Response · Authors · 2025-11-23
> **Official Comment by Authors (Part 3/5)**
>
> >Otherwise, it is hard to justify the capability of the method on various datasets. (2). The reported scores for PhysDiff are pretty low compared to the results in their paper. For instance, the reported FID of PhysDiff is 0.433, while in your Table 1, it is 3.411. The difference is huge. Please explain, and justify if it is a fair comparison.
>
> The original PhysDiff paper [7] relies on Residual Force Control (RFC) [8], i.e., an auxiliary, non-physical force field applied at the humanoid’s root to compensate for dynamics mismatch and contact errors. As we state in Appendix C, our implementation instead follows the realistic setup from the original MetaMotivo paper [9], which does not use any residual forces: the low-level controller is a PD policy whose action space directly specifies normalized PD targets, together with explicit joint-range and torque limits chosen to produce physically plausible motion. This controller is not designed to accept additional root forces, so “turning residual force on” is not a simple toggle; it would require expanding the action space and retraining the low-level policy. Modern controllers such as MaskedMimic [10] and Perpetual Humanoid Control [11] are likewise designed to operate without such non-physical residual forces, making evaluation with residual force off the appropriate choice when using a modern imitation controller.
> All SimDiff training data and all baselines, including our PhysDiff reimplementation, are generated and evaluated under this common RF-off setup. Because both methods share the same simulator and torque-only controller, the comparison between SimDiff and PhysDiff in Table 1 is intrinsically fair, even though the absolute PhysDiff FID is not directly comparable to the RF-on value reported in the original PhysDiff paper [7].
>
> >(3). As the paper aims for a better way of incorporating the physical guidance, it should compare with more baselines in this direction. Current comparison is not enough.
>
> Extending the physics‑guided comparison to Reindiffuse and Morph is unfortunately not straightforward. These two works were constrained by availability and timing. Reindiffuse has not released its code, and Morph is concurrent work, as it is an ICCV 2025 paper submitted after the ICLR deadline. Therefore, we unfortunately could not include these baselines.

---

> ### Author Response · Authors · 2025-11-23
> **Official Comment by Authors (Part 4/5)**
>
> >Regarding the compositional generalization, the comparison with baselines is not presented.
>
> > (4). Section 5.4, “Generalisation to diverse environments”. Only qualitative results are provided. More comprehensive comparisons, both qualitative and quantitative, are needed.
>
> Thank you for this valuable suggestion. Quantifying compositional generalization in our setting is non‑trivial: even prior work on compositional behavior with diffusion models[12,13] typically relies on low‑dimensional or grid‑world–style visualizations rather than quantitative metrics in realistic physics environments. Nevertheless, we added a concrete quantitative evaluation that complements the qualitative results in Figure 5.
>
> Compositional generalization here refers to SimDiff's capability to accurately predict the combined effect of multiple physical conditions from motions trained on each condition independently. Consider the scenario shown in Figure 5, where lunar gravity and horizontal wind conditions are combined. Let $\Delta x$ denote the pelvis displacement under horizontal wind blowing in the x-direction alone (with lunar gravity and no y-direction wind), and $\Delta y$ denote the pelvis displacement under horizontal wind blowing in the y-direction alone (with lunar gravity and no x-direction wind). Then, $\Delta xy$ is the pelvis displacement under simultaneous horizontal winds in both the x- and y-directions. If SimDiff generalizes compositionally, we expect $\Delta xy$ to be well approximated by a linear combination of the independent effects $\Delta x + \Delta y$. We quantitatively measure this using the coefficient of determination ($R^2$), which indicates how closely the combined displacement ($\Delta xy$) matches this linear approximation ($\Delta x + \Delta y$).
>
> To quantitatively evaluate compositional generalization, we specifically analyzed and quantified the scenario presented in Figure 5. We measured pelvis displacement under different horizontal wind conditions combined with lunar gravity (g = -1.62 m/s²). The official MDM baseline was evaluated using textual descriptions explicitly specifying each condition combination, without separate physics parameters:
> - "a person jumps high from a standing position under lunar gravity (g=-1.62 m/s²) with no wind"
> - "a person jumps high from a standing position under lunar gravity (g=-1.62 m/s²) with horizontal wind wx=2.0 m/s² and wy=0.0 m/s²"
> - "a person jumps high from a standing position under lunar gravity (g=-1.62 m/s²) with horizontal wind wx=0.0 m/s² and wy=2.0 m/s²"
> - "a person jumps high from a standing position under lunar gravity (g=-1.62 m/s²) with diagonal wind wx=2.0 m/s² and wy=2.0 m/s²"
> The resulting $R^2$ values, which measure how closely the combined horizontal displacement ($\Delta xy$) matches the linear approximation ($\Delta x + \Delta y$), are summarized below:
> | Method         | $R^2$ (higher is better) |
> | -------------- | --------------------- |
> | MDM | 0.224                 |
> | SimDiff (ours) | **0.870**             |
>
> The substantially higher $R^2$ achieved by SimDiff quantitatively demonstrates our model's superior compositional generalization capability compared to the original MDM model.
>
>
> > Ablation studies are not sufficient, e.g., the guidance scale, the key components.
>
> The classifier-free guidance scale is not the key component for controlling physics. We control the influence of the physical parameters using the adapter scale. We provided a detailed ablation study on this specific adapter scale in Table 2 of Appendix D. In addition, the effect of guidance scale itself has already been thoroughly ablated in the original MDM paper [14], so we follow their standard setting here.

---

> > ### Author Response · Authors · 2025-11-23
> > **Official Comment by Authors (Part 5/5)**
> >
> > [1] Christopher J K, Baek S, Fioretto F. Constrained synthesis with projected diffusion models[J]. Advances in Neural Information Processing Systems, 2024, 37: 89307–89333.
> >
> > [2] Li Z, Luo M, Hou R, et al. Morph: A Motion-free Physics Optimization Framework for Human Motion Generation[C]// Proceedings of the IEEE/CVF International Conference on Computer Vision (ICCV). IEEE, 2025: 14580–14589.
> >
> > [3] Han G, Liang M, Tang J, et al. ReinDiffuse: Crafting Physically Plausible Motions with Reinforced Diffusion Model[C]// 2025 IEEE/CVF Winter Conference on Applications of Computer Vision (WACV). IEEE, 2025: 2218–2227.
> >
> > [4] Baldassari L, Siahkoohi A, Garnier J, Solna K, de Hoop M V. Conditional Score-Based Diffusion Models for Bayesian Inference in Infinite Dimensions[J]. Advances in Neural Information Processing Systems, 2023, 36.
> >
> > [5] Cohan S, Tevet G, Reda D, Peng X B, van de Panne M. Flexible Motion In-betweening with Diffusion Models[C]// SIGGRAPH Conference Papers ’24 (ACM SIGGRAPH 2024). ACM, 2024.
> >
> > [6] Karunratanakul K, Preechakul K, Aksan E, Beeler T, Suwajanakorn S, Tang S. DNO: Optimizing Diffusion Noise Can Serve As Universal Motion Priors[C]// Proceedings of the IEEE/CVF Conference on Computer Vision and Pattern Recognition (CVPR). 2024.
> >
> > [7] Yuan Y, Song J, Iqbal U, Vahdat A, Kautz J. PhysDiff: Physics-Guided Human Motion Diffusion Model[C]// Proceedings of the IEEE/CVF International Conference on Computer Vision (ICCV). 2023: 15964–15975.
> >
> > [8] Yuan Y, Kitani K. Residual Force Control for Agile Human Behavior Imitation and Extended Motion Synthesis[C]// Advances in Neural Information Processing Systems (NeurIPS). 2020, 33.
> >
> > [9] Tirinzoni A, Touati A, Farebrother J, Guzek M, Kanervisto A, Xu Y, Lazaric A, Pirotta M. Zero-Shot Whole-Body Humanoid Control via Behavioral Foundation Models[J]. arXiv preprint arXiv:2504.11054, 2025.
> >
> > [10] Tessler C, Guo Y, Nabati O, Chechik G, Peng X B. MaskedMimic: Unified Physics-Based Character Control Through Masked Motion Inpainting[J]. ACM Transactions on Graphics, 2024, 43(6): 209.
> >
> > [11] Luo Z, Cao J, Winkler A, Kitani K, Xu W. Perpetual Humanoid Control for Real-time Simulated Avatars[C]// Proceedings of the IEEE/CVF International Conference on Computer Vision (ICCV). 2023: 10861–10870.
> >
> > [12] Janner M, Du Y, Tenenbaum J B, Levine S. Planning with Diffusion for Flexible Behavior Synthesis[C]// Proceedings of the 39th International Conference on Machine Learning (ICML). 2022.
> >
> > [13] Ajay A, Du Y, Gupta A, Tenenbaum J B, Jaakkola T S, Agrawal P. Is Conditional Generative Modeling All You Need for Decision-Making?[C]// International Conference on Learning Representations (ICLR). 2023.
> >
> > [14] Tevet G, Raab S, Gordon B, et al. Human Motion Diffusion Model[C]//The Eleventh International Conference on Learning Representations (ICLR). 2023.

---

### Author Response · Authors · 2025-12-04
**Summary of Rebuttal**

Dear All,

We sincerely appreciate the constructive feedback from the reviewers. We proposed SimDiff, which bridges projection and guidance to amortize physics constraints into the diffusion process. We are encouraged that reviewers agreed that this architectural choice leads to significant computational gains and efficiency compared to simulator-in-the-loop methods.

That said, reviewers also identified several important opportunities for improvement, addressed as follows (more details and updates in individual replies):
- **Clarified Theoretical Positioning (Resolving gF3b):** Reviewer gF3b questioned the strength of our original claim relative to prior projection-based or constrained diffusion methods. We acknowledged this and revised a sentence in Section 1 to explicitly position SimDiff as a "standalone physics-aware text-to-motion generative model that internalises simulator behaviour". We further explained, from a distributional viewpoint, that directly learning an environment‑conditioned score allows SimDiff to more closely approximate the desired physics‑constrained distribution than enforcing constraints via per‑step projections.
- **Validated Compositional Generalization (Resolving gF3b, ZHbu):** Reviewers requested quantitative proof for our generalization claims beyond visual examples. In response, we did an evaluation based on the coefficient of determination $R^2$ for pelvis displacement under lunar gravity plus planar wind. SimDiff achieves $R^2 = 0.870$, whereas an MDM, even when explicitly prompted with the physics conditions, achieves only $R^2 = 0.224$. This provides quantitative support that SimDiff internalizes the effect of physics well enough to handle unseen combinations such as low gravity plus diagonal wind, and we explicitly scope our claim to global scalar parameters in this simulated setting.
- **Established Baseline Fairness (Resolving gF3b, ZHbu, vsFh):** We addressed concerns regarding the performance gap with PhysDiff by clarifying the simulation configuration. We explained that our evaluation uses the Residual Force-off setting, which is the standard for modern, realistic imitation controllers. Under this shared setup, we argue that the comparison is fair and consistent with the latest literature, and that SimDiff offers a much better fidelity-efficiency trade‑off.

Since reviewers had not yet responded to our rebuttal before the OpenReview incident restricted further engagement, we were unable to receive their feedback. However, as outlined above, we have addressed all of their main concerns as carefully as possible, and we hope you will take these responses and new results into account when interpreting the original reviews and forming your recommendation.

Thank you for your time and feedback,

The authors of SimDiff

---

### Meta-Review · Area_Chair_4Fsy · 2025-12-28

**Summary:**

The authors present an additional branch for pre-trained text-to-motion models that allow conditioning on physical parameters. The architecture is trained on data obtained from simulations. The idea is straight-forward and makes sense, also the potential improvements in inference speed are very welcome, as appreciated by all reviewers after discussion. All reviewers started off with negatively leaning scores pre-discussion though, mainly due to concerns and questions regarding experiments (see below). Some concerns have been resolved in the discussion. However, main concerns about slim and unconvincing experiments where not sufficiently addressed in my opinion.

This paper is clearly in a borderline state. I think the general direction of including physical parameters as conditioning is very welcome, also pairing it with training data coming out of simulators. However, since the architectural innovation is a little bit limited (essentially a ControlNet-like branch for MDM), a paper that introduces this paradigm for physical plausibility needs to come with a large set of experiments that show strong results (1) against multiple baselines, (2) on multiple baseline architectures, (3) more than three physical parameters, (4) more convincing examples. Since the authors do not sufficiently bolster the work in this direction during discussion, I doubt the reviewers impression would have changed enough during discussion. I recommend to reject the work and suggest that the authors improve on the experiments and resubmit to a future venue.

**Reviewer Concerns:**

*1) Unclear positioning.* These concerns have been largely resolved the authors in the answers. The positioning has become clear but should be made more concise also in the paper. In my opinion the sentence "We focus on directly learning the conditional distribution from data. Instead of relying on inference-time substitutions, we train our model on simulator-generated data using classifier-free guidance" (Sec. 4.1) is an important sentence that could clarify the method early on.

*2) Missing related work.* Sufficiently addressed. I agree with the authors that it is not absolutely necessary that all human motion generation works are discussed. Their improvements are largely orthogonal and this work is mainly about how to condition them on additional physical environment parameters.

*3) Overstatements.* Sufficiently addressed, the introduction has been made more precise.

*4) Compositional generalization not supported by experiments.* Partially addressed, the authors have added one experiment in the answers. However, given how strong this claim is in the introduction, I would expect more exhaustive evaluation here.

*5) Only one dataset evaluated.* Not addressed but the used dataset is a very common dataset in this domain and could be sufficient.

*6) Questions regarding FID of PhysDiff.* Addressed and explained via different experimental setup parameters, which is reasonable.

*7) Narrow conditioning scope in experiments.* Not addressed. In my opinion, this is very crucial, as the method needs to be trained for all combinations of parameters that are expected to be used during inference. Having only 3 conditioning parameters is a strong limitation in this case.

*8) Selection bias.* Failed instances are discarded from the generated training set. This was not addressed by the authors. It remains unclear what effect his has. An ablation study is needed.

*9) Ablation studies.* Reviewers asked for additional ablation studies, which was not addressed by the authors.

*11) General slim and unconvincing evaluation.* The authors answered to this remark by justifying the comparison against PhysDiff, which is reasonable. However, I feel like the general impression is not resolved, given the limited setup of only 3 variables, a single dataset and a single evaluated architecture with MDM. The authors could setup their experiments more large scale, to show generality of their contribution.

**Reviewer Scores:**

All three reviewers had an initial score of 4. There where multiple questions regarding clarity, which were sufficiently answered and could have led to one or two reviewers increasing their score to 6. However, many concerns regarding insufficient evaluation remained unaddressed, which were important points for all three reviewers. Therefore, I doubt the average would have been increased to a value significantly above borderline.

---

### Decision · Program_Chairs · 2026-01-26

Reject